# Underlying Mechanisms behind the Brain–Gut–Liver Axis and Metabolic-Associated Fatty Liver Disease (MAFLD): An Update

**DOI:** 10.3390/ijms25073694

**Published:** 2024-03-26

**Authors:** Júlia Pauli De Cól, Enzo Pereira de Lima, Fernanda Moris Pompeu, Adriano Cressoni Araújo, Ricardo de Alvares Goulart, Marcelo Dib Bechara, Lucas Fornari Laurindo, Nahum Méndez-Sánchez, Sandra Maria Barbalho

**Affiliations:** 1Department of Biochemistry and Pharmacology, School of Medicine, Universidade de Marília (UNIMAR), São Paulo 17525-902, Brazil; juliapaulidecol@hotmail.com (J.P.D.C.); marcelo.dib.bechara@outlook.com (M.D.B.); 2Postgraduate Program in Structural and Functional Interactions in Rehabilitation, School of Medicine, Universidade de Marília (UNIMAR), São Paulo 17525-902, Brazil; ricardogoulartmed@hotmail.com; 3Department of Biochemistry and Pharmacology, School of Medicine, Faculdade de Medicina de Marília (FAMEMA), Marília, São Paulo 17519-080, Brazil; lucasffffor@gmail.com; 4Liver Research Unit, Medica Sur Clinic & Foundation, Mexico City 14050, Mexico; nmendez@medicasur.org.mx; 5Faculty of Medicine, National Autonomous University of Mexico, Mexico City 04510, Mexico; 6Department of Biochemistry and Nutrition, School of Food and Technology of Marília (FATEC), São Paulo 17500-000, Brazil

**Keywords:** brain–gut–liver axis, metabolic-associated fatty liver disease, MAFLD, insulin resistance, diabetes, dyslipidemia, obesity, organokines

## Abstract

Metabolic-associated fatty liver disease (MAFLD) includes several metabolic dysfunctions caused by dysregulation in the brain–gut–liver axis and, consequently, increases cardiovascular risks and fatty liver dysfunction. In MAFLD, type 2 diabetes mellitus, obesity, and metabolic syndrome are frequently present; these conditions are related to liver lipogenesis and systemic inflammation. This study aimed to review the connection between the brain–gut–liver axis and MAFLD. The inflammatory process, cellular alterations in hepatocytes and stellate cells, hypercaloric diet, and sedentarism aggravate the prognosis of patients with MAFLD. Thus, to understand the modulation of the physiopathology of MAFLD, it is necessary to include the organokines involved in this process (adipokines, myokines, osteokines, and hepatokines) and their clinical relevance to project future perspectives of this condition and bring to light new possibilities in therapeutic approaches. Adipokines are responsible for the activation of distinct cellular signaling in different tissues, such as insulin and pro-inflammatory cytokines, which is important for balancing substances to avoid MAFLD and its progression. Myokines improve the quantity and quality of adipose tissues, contributing to avoiding the development of MAFLD. Finally, hepatokines are decisive in improving or not improving the progression of this disease through the regulation of pro-inflammatory and anti-inflammatory organokines.

## 1. Introduction

Following the First and Second World Wars, the population’s nutritional intake changed and, consequently, consumption of ultra-processed food increased as did sedentary lifestyles; both were responsible for a decline in health and quality of life. Such a decrease in nutritional quality directly affects bacterial intestinal metabolism and its local microbiota, leading to metabolic conditions such as obesity, diabetes, and heart and liver diseases [1,2,3]. 

In parallel, emerging scientific knowledge concerning health and disease modulation by the microbiota and its several metabolites is focused on the so-called “brain–gut axis”; thus, the interruption of its healthy and effective communication results in a number of diseases associated with inflammation, cardiovascular risks, obesity, type 2 diabetes (DM2), and other endocrine pathologies, such as hypothyroidism and polycystic ovary syndrome [3,4,5,6,7,8,9,10].

The gut microbiota has a crucial role in the metabolism of substrates, including carbohydrates, proteins, polyphenols, and vitamins, thus influencing systemic homeostasis. The brain–gut axis includes a bidirectional communication system based on the information exchange between the central nervous system (CNS) and the gut. The “term” brain–gut–liver axis is related to the bidirectional communication system coming from the central nervous system, which forwards efferent signals to the gut lumen. The afferent ones in the gut are forwarded through the autonomic nervous system to the central nervous system. Direct neuronal signals and hormonal release perform the central control of the gut. It is also mediated by food intake, gut motility, liver glucose metabolism, and inflammatory and immune responses. On the other hand, the gut microbiome can affect neural physiology thanks to the production of lipopolysaccharides [11,12].

This complex network involves the neural, endocrine, and immune systems [13,14,15,16,17]. The intricate information exchanged between the CNS and the gut microbiome is mediated by microbiome-derived products, including serotonin, dopamine, norepinephrine, short-chain fatty acids (SCFAs), glutamate, γ-aminobutyric acid (GABA), secondary bile acids, tryptophan metabolites, and histamine. Short-chain fatty acids (SCFAs) mainly include propionate, acetate, and butyrate. These molecules are microbial metabolites synthesized by the bacterial fermentation of dietary fibers. SCFAs primarily work as energy for colonocytes. Researchers have shown that these molecules have an important role in the brain since their levels are modified in many neurological conditions such as multiple sclerosis, Alzheimer’s disease, Parkinson´s disease, epilepsy, and mood disorders [18]. The microbiome derivatives can be pivotal in modulating physiological and neurological mechanisms. On the other hand, communication from the brain–gastrointestinal tract is mediated by the autonomic nervous system, the vagal nerve, and the hypothalamic–pituitary–adrenal axis [19,20]. The new concept concerning the gut–brain–liver axis relates the composition of the small gut, gut vagus nerve, hypothalamus, and hepatic vagus nerve. Some of the gut microbiota factors are associated with the direction of the neural and hormonal complex, which informs the cephalic and hepatic center of the physiological changes [20,21,22].

The gut–brain–liver axis has received considerable attention because it also acts as the first line of defense against serum nutrient changes and metabolic disorders [19,23]. Thus, the hypothalamus detects hormonal levels, such as the presence of insulin, leptin, and glucagon-like peptide-1 (GLP-1), as well as nutritional, glucose, and fatty acids, regulating hepatic glucose metabolism through the vagal efferent nerve and, through the neural pathway of the gut–brain–liver axis, the control of the pathogenesis of MAFLD [24]. Concerning the hypothalamic axis, it is worth highlighting GLP-1, a prohormone secreted by L cells of the distal gut in the presence of glucose. This hormone stimulates insulin secretion by the beta cells of the pancreatic islets and signals insulin action by peripheral tissues to the CNS. Consequently, there is a decrease in appetite as an indication of satiety, activating the G protein-coupled receptor (GPR) GLP-1R. This connection controls plasma triglyceride and hepatic lipid levels as a hepatocyte protective complex [25,26].

In developed countries, metabolic-associated fatty liver disease (MAFLD) is considered one of the major causes of chronic liver disease and has a high incidence rate. The growing estimated global prevalence of MAFLD/MASLD is about 25–30%, resulting from the escalating epidemics of obesity and DM2 [19,27]. However, according to some data, 3% to 6% of Americans have Nonalcoholic Steatohepatitis (NASH), with higher prevalence in individuals with previously established metabolic diseases and 20% progress to liver cirrhosis [5]. Likewise, the metabolic dysfunction associated with MAFLD is a frequent cause of chronic liver diseases and possibly cirrhosis, liver carcinoma, and liver transplant. Therefore, clinically, three basic criteria are evaluated for diagnosis as follows: overweight or obesity, the presence of DM2, or evidence of metabolic disorders [9,28,29]. Metabolic syndrome is characterized, according to the National Cholesterol Education Program Adult Treatment Panel III (NCEP ATP III) criteria, by the presence of three to five risk factors that include elevated abdominal circumference, triglyceride, blood pressure, and fasting glycemia and reduced High-density Lipoprotein-cholesterol (HDL-c) levels (Third Report of the National Cholesterol Education Program (NCEP) Expert Panel on Detection, Evaluation, and Treatment of High Blood Cholesterol in Adults (Adult Treatment Panel III final report, 2002)) [30]. The condition loads its key point on insulin resistance, inducing liver lipogenesis and further complications. Lipid accumulation increases oxidative stress and liver inflammation, potentialized by gene inducers, gut microbiota dysfunctions, and liver metabolism disruption [31,32,33,34].

Thus, a challenge in diagnosing MAFLD is to determine if metabolic syndrome leads to a deficit in the production of adipose tissue hormones. For example, gut peptides such as ghrelin, cholecystokinin (CCK), and GLP-1 may activate the afferent vagus nerve and stimulate the CNS; such signs are integrated and, subsequently, regulate food intake and the metabolism of liver glycolipids through a negative feedback cycle [15,21,26,30,35]. 

Therefore, the indissociation between obesity, DM2, metabolic syndrome, MAFLD, and the brain–gut–liver axis is undeniable. This complex and not fully understood relationship is closely linked to the risk of hepatic fibrosis, which is aggravated with advancing age [36,37]. 

In this sense, the higher the severity to which the liver is subjected, and the vulnerabilities caused by aging, the greater the cognitive problems that the patient may face. Moreover, insulin is a protein that, upon crossing the blood–brain barrier (BBB) and leading to a hyperglycemic state associated with insulin resistance, results in harmful effects on brain function, involving advanced glycation end products, oxidative stress, and glucose-induced neurotoxicity [38,39]. However, in severe hypoglycemic conditions, it also leads to cognitive impairment by downregulating insulin receptors at the BBB, impairing its transport in this region. When the accumulation of pro-inflammatory factors occurs in the bloodstream, through active transport or directly in the circumventricular areas, unique points of communication in the bloodstream, such factors can penetrate the brain tissue and cerebrospinal fluid (CSF). Once damage-associated molecular patterns (DAMPs) and pro-inflammatory cytokines reach the CNS, they activate microglia, triggering a complex immune response in the CNS that promotes toxicity, chronic inflammation, and cognitive decline [19,40]. 

In view of the above, this comprehensive review aimed to show the connections between the brain–gut–liver axis and MAFLD. Furthermore, we discuss the role of adipokines, myokines, and hepatokines and clinical advances regarding this liver condition.

## 2. Metabolic Repercussions in the Imbalance of the Brain–Gut–Liver Axis

### 2.1. Metabolic Associated Fatty Liver Disease (MAFLD)

The recent modification of the term NAFLD (Nonalcoholic Fatty Liver Disease) to MAFLD is aimed at enhancing precision and broadening the coverage of the pathological disease spectrum. Consequently, it leads to improvements in the treatment and prognosis of patients, encompassing not only those in nonalcoholic contexts but also all individuals meeting the criteria for fatty liver dysfunction. This modification correlates them with other pathologies associated with metabolic deregulation, thereby avoiding pre-established statements [41]. Furthermore, the MAFLD definition is better able to identify hepatic fibrosis compared with NAFLD [42]. For instance, the obesogenic microbiome—characterized by metabolic dysfunctions—potentializes the development of fatty liver diseases such as MAFLD [28,34,43]. Recently, another proposal regarding nomenclature aimed to revise the MAFLD definition to include at least one of five cardiometabolic risk factors. The newly proposed term is Metabolic Dysfunction-associated Steatotic Liver Disease (MASLD) [44]. However, a recent study provided evidence that MAFLD better identifies patients at a higher risk of liver fibrosis and progression of the condition [45]. Therefore, as we believe MAFLD is more appropriate, we will continue to use this term throughout the text.

Figure 1 summarizes the differences between NAFLD and MAFLD.

### 2.2. Imbalance in the Liver Metabolic Profile

The liver plays a crucial role in producing inflammatory biomarkers in response to insulin resistance, oxidative stress, and a pro-inflammatory state [46,47]. Therefore, the organ is pivotal in the pathogenesis of cardiovascular diseases (CVDs) during the necroinflammatory stage. In this scenario, the hepatic release of several potential factors tightens the relationship between MAFLD and cardiovascular risks, such as systemic inflammation, altered lipid metabolism, oxidative stress, and prothrombotic state [48,49,50]. In patients with DM2 who exhibit subclinical diastolic dysfunction of the left ventricle, alongside normal systolic function and no history of acute myocardial infarction (AMI), there appears to be an association between MAFLD and the ectopic accumulation of fat within the myocardium. In certain cases, this interrelationship is facilitated by the early onset of atherogenic dyslipidemia. This pro-fibrinogenic environment directly influences cholesterol metabolism in gut dysbiosis [51], consequently reducing serum levels of HDL cholesterol while elevating plasma concentrations of triglycerides and low-density lipoprotein cholesterol (LDL cholesterol). These changes are driven by the hepatic accumulation of free fatty acids (FFAs), leading to increased secretion of very low-density lipoprotein cholesterol (VLDL cholesterol). The results are associated with elevated levels of C-reactive protein (CRP) and Plasminogen activator inhibitor-1(PAI-1), which are mediated by Interleukin (IL)-6, Tumor Necrosis Factor-α (TNF-alpha), and Pentraxin-3 (an acute inflammatory protein with pro-coagulant actions). Gut dysbiosis, driven by elevated levels of Trimethylamine N-oxide (TMAO), increases cardiovascular risk by reducing the concentration of enzymes involved in reverse cholesterol transport and bile acid synthesis. This dysregulation of the gut microbiota is associated with the progression of atherosclerosis [52,53,54]. Several mechanisms linking adiposity and MAFLD to carcinogenic development include alterations in insulin metabolism, the bioavailability of insulin-like growth factor 1 (IGF-1), adipokine pathophysiology (such as adiponectin, leptin, and resistin), and increased propensity for systemic inflammation. This creates an environment conducive to the emergence of neoplasms, particularly colorectal cancer [41,55,56,57,58,59].

### 2.3. Biochemical and Cellular Patterns of MAFLD

The hepatic cellular composition responsible for lipid uptake, synthesis, oxidation and distribution to peripheral tissues includes parenchymal cells, which represent 78% of its total cell population, hepatocytes and non-parenchymal cells, such as sinusoidal endothelial cells (LSECs—liver sinusoidal endothelial cells), Kupffer cells (KCs, resident macrophages activated by liver injury), stellate cells (HSCs, hepatic stellate cells), and Natural killer (NK) cells. Cytokines (TNF-α), Interleukins (IL-1, IL-6, IL-12, IL-18), and inflammatory chemokines released by KCs contribute to the pathogenesis of MAFLD, a process mediated by pro-inflammatory KC M1 and anti-inflammatory KC M2 cells [55]. HSCs, in response to the inflammatory process, suffer possible damage caused by lipotoxicity and transform into myofibroblasts, secreting collagen and developing points of fibrosis. Fibrosis occurs as a reparative reaction to prolonged injuries, diminishing the ability for regeneration by causing an overabundance of substances produced by fibroblasts, including collagen and extracellular matrix components. This process may advance to the development of liver cirrhosis [25]. On a molecular level, the hepatic necroinflammatory stage is propelled by heightened synthesis of Platelet-Derived Growth Factor (PDGF), a powerful stimulant for mesenchymal cell proliferation, as well as Transforming Growth Factor-β (TGF-β), which initiates cellular proliferation. Additionally, TNF-α and IL-6, released as components of the inflammatory response, contribute to fibrosis induction [43,60,61].

Furthermore, PDGF regulates the expression of metallopeptidases (collagenases) (MMP2 and MMP9) and metallopeptidase inhibitor 1 (TIMP1). By inhibiting collagenase activity, it amplifies the accumulation of the extracellular matrix (ECM), thereby impeding its breakdown. Moreover, PDGF triggers various signaling pathways including mitogen-activated protein kinase (MAPK), c-Jun N-terminal kinase (JNK), and protein kinase B. MAPK signaling and its activation foster oxidative stress by modulating nuclear factor E2-related factor 2 (Nrf2) and nuclear factor kappa B (NF-κB), which are both implicated in liver disorders. When MAPK signaling coincides with a high-fat diet, it triggers lipid accumulation, leading to the subsequent generation of inflammation and ROS. Consequently, the inflammatory reactions stemming from MAPK signaling worsen the advancement of MAFLD [62,63]. Similarly, the activation of MAPK, and JNK activation, occurs within the liver in response to nutritional and metabolic stress. The aberrant activation of MAPKs, particularly JNKs, induces detrimental alterations in hepatocytes, exacerbating the condition of MAFLD. Hepatocytes express MAPKs that transmit extracellular and intracellular signals, governing processes like proliferation, differentiation, apoptosis, and cellular metabolism. Consequently, the heightened activation of the metabolic stress-induced MAPK cascade in hepatocytes causes significant damage to the liver. This process involves upstream MAPK kinases (MAP3K), including apoptosis signal-regulating kinase 1 (ASK1), mixed-lineage kinases (MLKs), TAK1, MAPK kinases (MAP2K) such as mitogen-activated protein kinase kinase 4 (MKK4), MKK7, and terminal stress kinases like JNK and p38 [63,64]. Moreover, serving as an inflammatory pathway associated with various chemical mediators, MAPK activation can impact other bodily systems, including the cardiovascular system. For instance, p38δ MAPK modulates the activation of the Nod-like receptor protein 3 (NLRP3) inflammasome, exhibiting heightened expression in coronary atherogenesis. This, in turn, results in the dysfunction of cardiomyocytes and triggers an inflammatory response in vascular smooth muscle [65,66].

TGF-β is an important marker of fibrosis in the liver and is produced by HSCs, KCs, LSECs, and activated hepatocytes. Like PDGF, TGF-β1 controls the expression of genes responsible for matrix production and hampers their breakdown. Additionally, it triggers apoptosis in hepatocytes within fibrotic livers, contributing to tissue loss and subsequent organ shrinkage. TNF-α is another pro-inflammatory mediator that HSCs, KCs, monocytes, and macrophages produce. In the context of liver fibrosis, TNF-α fosters excessive ECM production in the liver, while IL-1 prompts the activation of HSCs, initiating ECM synthesis. Consequently, all these mediators collectively contribute to the progression of liver fibrosis, cirrhosis, and eventual impairment of liver function. Concerning LSECs, their exposure to lipotoxicity diminishes the levels of nitric oxide (NO) and elevates the presence of ROS. This imbalance leads to oxidative stress, heightening the likelihood of developing NASH [67,68].

An excess of ROS triggers mitochondrial dysfunction, alters nucleic acids, and promotes an accumulation of advanced glycation end products (AGEs). These processes culminate in apoptosis, facilitated by the activation of caspase 3 and 9. ROS activity further stimulates the release of both pro-inflammatory and anti-inflammatory cytokines and organokines, including IL-6, TNF-α, leptin, and adiponectin. These molecules play roles in hepatic inflammation and may potentially induce changes in genes associated with neoplastic transformation [55,69]. 

### 2.4. Oxidative Stress and MAFLD

Alongside cellular changes, high-calorie diets, often associated with a sedentary lifestyle, are pivotal factors contributing to the development of MAFLD. The pathophysiology of the disease centers around the accumulation of fat in the liver, leading to an imbalance between the influx of fatty acids from the diet to the liver, lipogenesis, and lipolysis of adipose tissue. This imbalance is closely associated with the synthesis and oxidation of lipids and triglycerides, which are subsequently exported from the liver in the form of VLDL-c. Hence, during the initial phases of MAFLD, there is an increase in the secretion of VLDL-c and lipid beta-oxidation. In individuals with MAFLD, 15% of the fatty acids within the liver originate from dietary sources, 59% from circulation, and 26% from lipogenesis [56,70].

Lipogenesis primarily converts surplus carbohydrates, particularly glucose, into fatty acids. VLDL secretion and the transportation of fatty acids via apoprotein B100 are associated with endoplasmic reticulum stress and progression to MAFLD. This transport to the Golgi complex is facilitated by specific vesicles and protein components, notably the transmembrane 6 superfamily 2 (TM6SF2), which plays a direct role in the development of MAFLD [71,72]. In this context, hyperinsulinemia exacerbated by insulin resistance impedes hepatic lipid export, leading to the degradation of ApoB100 and suppression of MTTP synthesis (mitochondrial triglyceride transfer protein). MTTP is responsible for transferring lipid components such as cholesterol esters, triglycerides, and phospholipids to the endoplasmic reticulum lumen, where these lipoproteins are assembled. In the clinical progression of MAFLD, insulin resistance, which is prevalent in obese patients, promotes lipogenesis without a concomitant reduction in VLDL-c production. Consequently, researchers have sought agents capable of inhibiting MTP in the liver or the intestine. Liver-specific MTP inhibitors can decrease VLDL-c secretion, primarily composed of apoB100. Intestine-specific MTP inhibitors reduce the production of chylomicrons containing apoB48. Studies indicate that these inhibitors significantly reduce total cholesterol, LDL-c, VLDL-c, triglycerides, and ApoB levels, while also exhibiting insulin-sensitizing and anti-atherosclerotic effects [56,61,73,74,75,76].

As mentioned earlier, factors related to mitochondrial dysfunction play a significant role in the progression of MAFLD, These factors include reduced beta-oxidation, leading to impaired ATP depletion, and an increased production of ROS due to the uncontrolled flow of fatty acids. This results in chronic production of Acetyl-CoA and dysregulation of the Krebs cycle. Moreover, mitochondrial structural changes are stimulated, further exacerbating lipid accumulation in hepatocytes [55]. Excessive fat accumulation can lead to a toxic cascade, hindering lipid droplet remodeling, exacerbating the clinical condition, and resulting in hepatocyte death through several mechanisms. These mechanisms include apoptosis via the activation of caspase 3 and 9, as well as necrosis, necroptosis, pyroptosis, and ferroptosis [60,77]. 

Mitochondria, which are essential for ATP synthesis through the metabolism of pyruvic acid and fatty acids, also contribute to the generation of ROS during aerobic reactions. In a detrimental cycle, excessive lipid accumulation within the liver cells stimulates mitochondrial fatty acid oxidation and ROS production. Consequently, as mitochondrial function declines, the oxidative impact of metabolic reactions intensifies, leading to increased production of reactive species and damage to mitochondrial constituents, including membranes, proteins, and genetic material. In response to oxidative stress, mitochondria initiate their quality control mechanisms (MQC) to preserve their architecture and genetic integrity, thereby impeding the progression of MAFLD [78,79]. The interplay between endoplasmic reticulum stress (ERS) and oxidative stress significantly contributes to the pathogenesis of this condition [80]. Excessive ROS also triggers the secretion of inflammatory cytokines, including IL-6, TNF-α, and leptin, which further exacerbate liver inflammation and potentially contribute to the development of carcinomas [81]. IL-6 serves as a pivotal signal in ROS-induced liver injury, promoting cell proliferation and activating antiapoptotic pathways via the STAT3 signaling pathway, which is known for its oncogenic properties. Simultaneously, TNF-α contributes to disease progression and hepatocarcinogenesis by activating the JAK2/STAT pathway [43,60,82,83]. 

Significantly, mitochondria-associated membranes (MAMs) serve as structural connections facilitating the clustering of molecules, particularly for calcium ion exchange, lipid transfer, and ROS [84,85]. Consequently, research has investigated the pivotal roles of reactive nitrogen species (RNS) and ROS in MAFLD, demonstrating that the intactness of MAMs is crucial for normal communication between the endoplasmic reticulum and mitochondria. Disruption of MAM integrity leads to communication breakdown, directly or indirectly causing disturbances in calcium ion homeostasis, thereby increasing stress on the endoplasmic reticulum and oxidative conditions. Consequently, MAM involvement in glucose and lipid metabolism, chronic inflammation, and insulin resistance in MAFLD is underscored. Furthermore, alterations in the physiological function of these structures, particularly in phospholipid trafficking, potentially exacerbate the pro-inflammatory environment in metabolic diseases like MAFLD [21,77,86]. Table 1 provides a summary of the sources and effects of ROS and RNS on MAFLD.

### 2.5. Brain–Gut–Liver Axis

As previously mentioned, an imbalance in the microbiome disrupts the intricate interplay between the brain, gut, and liver. Several studies have demonstrated a correlation between chronic psychological stress and liver injury, including fibrosis. This process involves alterations in the diversity of the gut microbiome and increased intestinal permeability induced by psychological stress. Lipopolysaccharides (LPSs) reaching liver tissue can activate toll-like receptor-4 (TLR-4), highlighting the existence of the brain–gut–liver axis [87]. Additionally, MAFLD itself increases the risk of neurodegeneration. Disruptions in the axis can lead to hepatic encephalopathy, a neuropsychiatric condition associated with acute liver failure and cirrhosis [11,88,89,90].

Chemical signals from the environment are transmitted throughout the enteric nervous system, including vagal sensory nerves and sympathetic neurons. These nerves directly or indirectly perceive the intestinal microenvironment and convert these chemical signals into nerve impulses, which are propagated throughout the intestine and the CNS [19]. The hypothalamic–pituitary–adrenal (HPA) axis also plays a crucial role in endocrine pathways associated with the brain–gut–liver axis. In response to stressors, corticotropin-releasing hormone (CRH) and adrenocorticotropic hormone (ACTH) are released [91,92]. ACTH stimulates cortisol production, along with other stress-related substances that can impact various physiological pathways, including those in the gut. While cortisol primarily affects the brain, many intestinal cells express cortisol receptors, which can interfere with their function and the composition of the microbiome [93,94,95]. The brain–gut axis may include microbiome-stimulating CNS stress circuits [96].

The immune system plays an essential role in the brain–gut axis. The gut houses a robust immune system, and communication with the CNS involves signaling molecules like cytokines and chemokines. Additionally, products derived from the gut microbiome can modulate immune cell functions, potentially triggering local or systemic inflammation [97,98,99].

Medina-Julio [19] suggests that within the brain–gut–liver axis framework, hepatic tissue acts as the conductor of physiological responses, particularly concerning MAFLD and its impact on cognitive function. The portal vein serves as the vital link connecting the liver and the gut. The signals originating from the gut microbiome establish a direct link between liver health and the brain–gut axis. Imbalances within this interconnected system, as observed in liver diseases, can result in increased intestinal permeability. This, in turn, allows for the heightened translocation of harmful substances into the bloodstream [100]. This leaky gut condition, marked by heightened intestinal permeability and resulting in symptoms like chronic diarrhea, constipation, and bloating, contributes to systemic inflammation. Importantly, it may also impact cognitive functions [101]. MAFLD is also linked to various neurological manifestations, including reductions in brain volume, cognitive impairment, and cerebrovascular disease [102]. 

Figure 2 summarizes the mechanisms associated with the brain–gut–liver axis.

While the liver plays a crucial role in regulating the metabolism of potentially toxic compounds, the pathophysiology of chronic liver diseases, such as MAFLD, involves changes in bile acid profiles exacerbated by the Western diet (WD). These changes act as promoters of neurodegenerative disorders, driven by factors like lipotoxicity, mitochondrial dysfunction, endoplasmic reticulum stress, and oxidative stress, leading to the synthesis of ROS. This disruption compromises metabolic homeostasis and increases the concentration of toxic chemicals in metabolism. Consequently, due to the saturation of ROS production, susceptibility to neuroinflammation is observed, highlighting the intricate relationship between the liver and the brain [103].

In hepatodegenerative diseases, the degradation of amyloid-β (Aβ) peptide by the liver is compromised. This leads to the accumulation of alpha-synuclein amyloid fibrils, which, along with the ongoing inflammatory process, are implicated as primary causes of Alzheimer’s disease in genetically susceptible individuals. Additionally, Parkinson’s disease is triggered by the presence of Lewy bodies, composed of the overproduction of alpha-synuclein during the neuroinflammatory process. The reduction in lipoprotein receptor protein-1 (LRP-1), responsible for removing detoxified Aβ from liver circulation, leads to increased levels of IL-6 and circulating Aβ, as well as triglycerides, cholesterol, and transaminases. Moreover, the pathophysiology of these diseases is also supported by hyperammoninemia resulting from liver damage. Its accumulation contributes to the generation of free radicals involved in the neurochemical cascade of hepatic encephalopathy. Conversely, eating habits can be correlated with hepato-neurodegenerative pathologies. Food intake regulates the synthesis of protein hormones such as insulin and GLP-1. These hormones can activate signal transduction pathways in the hypothalamus and hippocampus, promoting synaptic activities associated with cognition and memory. In hyperglycemic and insulin-resistant individuals, there is a reduction in the production of brain-derived neurotrophic factor (BDNF) and IGF-1 due to high-calorie diets. This directly associates WD with the development of MAFLD, impairing the proper absorption of nutrients and vitamins in the intestine. Additionally, WD affects hippocampal learning functionality and memory due to the accumulation of fat and sugar, leading to increased concentration of pro-inflammatory lipids such as Aβ, which are correlated with neuro-pathologies [104].

Bile acids (BAs), synthesized by the liver through cytochrome P450 and cholesterol hydroxylation via CYP7A1, experience compromised metabolism and homeostasis during WD, exacerbating systemic inflammation. Consequently, due to their permeability to the blood–brain barrier, the activation of glucocorticoid receptors occurs, leading to the inhibition of hepatic glucocorticoid clearance and the hypothalamic–pituitary–adrenal axis [105]. Bile acids are amphiphilic steroid molecules originating from cholesterol in the liver and are crucial for digesting and absorbing fat-soluble substances in the intestine. Their amphiphilic nature allows them to act as surfactants, forming micelles with cholesterol, lipids, and fat-soluble vitamins, facilitating digestion and absorption. Primary bile acids, such as cholic acid and deoxycholic acid, are synthesized in hepatocytes from cholesterol. Secondary bile acids are produced from primary bile acids by intestinal bacteria. Higher levels of secondary bile acids and their conjugated forms, along with higher ratios of secondary to primary bile acids, are associated with worse markers of diseases like Alzheimer’s and Parkinson’s, including beta-amyloid, tau, neuropil, and impaired cognitive function. Conversely, lower levels of primary bile acids are linked to increased amyloid deposits in the brain, faster accumulation of lesions in white matter, and increased brain atrophy. These factors may contribute to the development of neurodegenerative diseases such as Alzheimer’s and Parkinson’s [106,107].

BAs have been found in the brain, indicating a possible connection between BA and neurological function and disease, as they are transported to the brain systemic circulation. BAs play a significant role in the microbiota–gut–brain axis, exerting anti-inflammatory, antioxidant, and neuroprotective activities in neurological diseases. However, with a diet rich in fat and cholesterol, this scenario can change, leading to damage to the human body. A high-fat and fructose diet (HFFD), common in Western societies, exacerbates disorders in gut microbiota and BA metabolism. This type of intake hampers both the formation of BAs and their full functioning [108].

Firstly, BAs serve as ligands for cell surface receptors including TGR5 and sphingosine-1-phosphate receptor 2, as well as muscarinic receptors M2 and M3, all of which are expressed in the brain. Additionally, they play a crucial role in the majority of cholesterol metabolism within this organ. Secondly, Alzheimer’s disease is marked by the buildup of β-amyloid peptide and tau protein within the brain. β-amyloid peptide originates from amyloid precursor protein (APP) through the actions of β and γ-secretases. In Alzheimer’s disease, the inhibition of autophagy for clearing this surplus protein is linked. Given that bile acids (BAs) play a role in this removal process, their regulation directly impacts the prognosis of the disease when adversely affected. Moreover, the malfunction of phosphatase and tensin homolog-induced kinase 1 (PTEN)-induced kinase 1 (PINK1) is recognized as a significant contributor to Parkinson’s disease, where their imbalance affects mitophagy, underscoring the vital role of mitochondrial quality control in mitigating Parkinson’s disease. Finally, bile acids (BAs) are also linked to autophagy and the maintenance of mitochondrial quality control [109].

Another mechanism associated with Alzheimer’s disease involves the influence of bile acids (BAs), wherein the buildup of cholesterol in the brain contributes to hepatic encephalopathy, a chronic liver ailment resulting in neuronal loss and heightened susceptibility to Alzheimer’s disease. This effect is mediated by bile acids’ interaction with the farnesoid X receptor. The pronounced accumulation of cholesterol due to dietary factors and impaired bile acid formation exacerbates this process [110,111]. 

TMAO is a metabolite of gut microbiota with connections to neurocognitive decline. It arises from the dietary intake of choline and carnitine. TMAO has emerged as a risk factor for Alzheimer’s disease because of its involvement in various pathophysiological pathways, including the aggregation of Aβ peptide and tau protein, which are central to the pathology of Alzheimer’s. Additionally, TMAO can activate astrocytes, trigger inflammatory responses, and may contribute to cognitive deterioration [112,113].

#### The Brain–Gut–Liver Axis in MAFLD

The liver and intestinal tract are anatomically and functionally related, as both originate from the same germinal region during embryonic development. Consequently, the liver serves as the primary line of defense against endotoxins produced by the intestinal tract. Changes in intestinal microbiota, such as the aforementioned leaky gut syndrome, can thus contribute to the development of MAFLD [114,115,116]. In parallel, metabolic dysregulations produced by pathologies such as MAFLD exacerbate both intrahepatic and systemic pro-inflammatory conditions, heightening the risk of neurodegeneration associated with hepatic encephalopathy. This debilitating neuropsychic condition is intrinsic to acute liver failure and cirrhosis. Consequently, it can be said that the connection axis between hepatic and intestinal functions is mediated autonomously by the nervous system and its neural connections through the vagus nerve. The neural pathway represents the regulatory direction of endogenous glucose production (EGP). Some studies suggest that during heightened systemic insulin levels, as observed in the postprandial state, the brain may directly regulate and suppress EGP. However, this regulatory mechanism becomes compromised with increased adiposity [11]. In patients with MAFLD, intestinal bacteria migrate through the portal vein to the liver and cause abnormal immune system activation, triggering inflammation and injury. Furthermore, interactions between the gut and liver are bidirectional. Inflammatory hepatokines disrupt the function of the intestinal mucosal barrier, causing fragmentation of the tight junctions within the intestinal epithelium. This process generates a detrimental liver–gut cycle during MAFLD. Additionally, patients with MAFLD exhibit increased intestinal permeability, further complicating the liver–gut relationship [77]. In this context, changes in the gut microbiota of these patients are primarily characterized by a decrease in bacterial diversity [22,117].

The gastrointestinal tract (GIT) is responsible for the majority of the production of hormones involved in the brain–gut–liver axis, with emphasis on ghrelin, CCK, peptide tyrosine–tyrosine (PYY), leptin, ghrelin, 5-hydroxytryptamine (5-HT), and GLP-1 (these peptides are secreted by enteroendocrine cells). The gut microbiota plays a crucial role in regulating the levels of these peptides, thereby extending its influence on the vagal afferent pathway [26,118]. GLP-1 has receptors in the hepatic portal region and neurons, suggesting its prolonged action from the liver to the walls of the intestinal loops. In healthy individuals with stable insulin and glucagon concentrations, GLP-1 inhibits EGP, mediated by a direct effect of GLP-1 on the liver or by a neuro-mediated inhibitory reflex. Thus, in patients with insulin resistance, the administration of Exenatide, a GLP-1 receptor agonist drug, reduces liver fat and liver enzymes, favoring the treatment of MAFLD [119,120]. Thus, this hormonal response is suppressed through a neuro-mediated inhibitory reflex in EGP, leading to a reduction in liver fat among individuals with insulin resistance. Furthermore, ghrelin, which is produced by the stomach and the duodenal portion of the gut, regulates food intake by stimulating appetite and inhibiting the action of CCK. Conversely, CCK is secreted by intestinal endocrine cells during feeding and binds to the following two main receptors: CCK-A, predominantly found in the GIT, and CCK-B, primarily located in the brain. This hormone acts in a complementary manner to insulin. Similar complementary effects have been observed with leptin, which binds to CCK-A receptors in the intestinal vagal fibers, transmitting signals to the cephalic trunk to signify the end of a meal [11]. 

The liver and GIT are physiologically bidirectional organs: in one direction, the hepatic store secretes bile and other bioactive mediators via the bile duct to the intestinal cavity, while, in the other, metabolic nutrients are directed to the liver via the portal vein after reabsorption in the gut. In the same way, intestinal bacteria and their products, such as short-chain fatty acids and LPSs, are also transported through the portal circulation, exposing the liver to the intestinal macroenvironment [77]. 

During the occurrence and development of MAFLD, fructose is another crucial dietary element, acting to damage the intestinal barrier and induce dysbiosis [121]. This hexose rapidly enters the liver through glucose transporter 2 (GLUT2) and, at the cellular level, can be converted into glucose and fatty acids. The lipogenic pathways become active after an acute fructose load [122,123,124]. Furthermore, an imbalance in the liver entry and metabolism of fructose leads to Acetyl-Coenzyme A production (glycolytic pathway), over the liver cells’ oxidative capacity, resulting in neolipogenesis mediated by the stimulation of Sterol Response Element Binding Protein 1c (SREBP1c) and Carbohydrate-Responsive Element-Binding Protein (ChREBP). The metabolic results of these processes are the increase in Fatty Acid Synthase (FASN) and Acetyl-CoA Carboxylase (ACC) expression, upregulating lipid synthesis. Overload of the glycolytic pathways also commences an accumulation of glycolysis intermediates, including glycerol-3-phosphate, captured in triglyceride synthesis (TG) [122,125]. Stimulation of neolipogenesis and accumulation of lipids in the liver can lead to increased hepatic insulin resistance. Additionally, fructose can inhibit adiponectin synthesis and release, further exacerbating insulin resistance and promoting liver steatosis. These events collectively elevate the production of oxidative species and establish an oxidative stress environment, which is associated with endoplasmic reticulum stress and mitochondrial dysfunction. These factors contribute to inflammatory processes and the progression from simple steatosis to NASH [43,122,126,127].

Intestinal dysbiosis causes an increase in the secretion of LPS—the central component of the outer membrane of Gram-negative bacteria—and endotoxin. Such bacterial products are directed to the hepatic store, physiologically, through the portal vein, relating to pathogenesis from MAFLD. In this aspect, studies have shown that plasma LPS-binding proteins in patients with MAFLD are significantly increased [128,129,130].

Therefore, LPS binds to LPS-binding proteins and then binds to TLR-4, triggering insulin resistance, followed by an immune response and greater propensity for inflammation; that is, LPS increases the levels of gut-derived TLR4 since this receptor is activated in hepatic Kupffer cells and stellate cells. Further pro-inflammatory and pro-fibrotic pathways are stimulated through several cytokines, including IL1, IL6, and TNF [77,131,132,133,134]. In MAFLD, TLR-4 signaling initiates local inflammatory changes and develops hepatic steatosis. TLR-4 downregulation in hepatocytes has been identified to resolve liver inflammation, improve insulin resistance, and reduce fat concentration in the liver [22].

Despite the role of the brain, the tenth pair of cranial nerves—the vagus nerve—plays a critical role in the communications of the BGL axis; thus, the enteric nervous system, associated with the synthesis of neurotransmitters such as acetylcholine, adrenaline, and serotonin (5-HT), mainly, promotes the regulation of hepatic lipid metabolism through the intestine–liver axis [11,42]. In this aspect, brain function in the axis is compromised during the onset of MAFLD, leading to extrahepatic complications through nerve dysfunction, brain injuries and changes in cerebral perfusion, acceleration of brain aging, and an increased risk of ischemic and hemorrhagic stroke, in addition to the oxidative scenario, elevated during pathological evolution, which promotes changes in mitochondrial function and structure and a consequent reduction in neuronal metabolism. In turn, metabolic dysfunctions in specific areas of the brain, i.e., the thalamus, hippocampus, and prefrontal cortex, can cause cognitive deficits since several metabolites such as N-acetyl aspartate, creatine, choline, glutamate, and taurine, involved in energy metabolism and the maintenance of brain functions, are impacted by the multisystem disease. In recent studies, patients with high average daily calorie intake have been found to have higher concentrations of inflammatory cytokines in the brain, resulting in microgliosis, astrocytosis, and neuronal damage [86,135,136]. Figure 3 shows the main events in the occurrence of MAFLD.

## 3. The Role of Organokines

Initially, for a satisfactory understanding of the complex interaction between the brain–gut–liver axis and its impairment in MAFLD, one must understand the extensive network of adipokines, myokines, osteokines, and hepatokines involved in the chemical and hormonal functioning of this organic communication.

### 3.1. Adipokines

Adipokines play a central role not only in the genesis of MAFLD but also in digestive diseases linked to obesity such as cholelithiasis, Barrett’s esophagus, and esophageal and colorectal cancer, as well as in pancreatic neoplasia and diabetes, with the two main adipokines being adiponectin and leptin, and, to a lesser extent, omentin, resistin, vaspin, and visfatin [56,137,138,139]. Furthermore, the expression of serum exosomal miRNAs can aid in detecting MALFD. Changes in the levels of liver enzymes, such as alanine transaminase (ALT), aspartate aminotransferase (AST), alkaline phosphatase (ALP), and Gamma Glutamyl Transpeptidase (GGT), are important not only for the differential diagnosis of liver disease but also for evaluating the severity of the disease [15,136,140].

Adiponectin, secreted into the systemic circulation by adipocytes (both white and brown adipose adipocytes), is responsible for the activation of distinct cellular signaling pathways in different tissues—it has an insulin-sensitizing action, that is, its predominant receptors function in the regulation of glucose and lipid metabolism—AdipoR1, abundant in striated skeletal muscle, and AdipoR2, the majority in liver tissue—favor increased sensitivity to insulin and, consequently, its hypoglycemic effect. In particular, in the liver, this function is exerted by AdipoR1, which is involved in the activation of AMP-activated kinase (AMPK), and by AdipoR2, which is involved in the activation of peroxisome proliferator-activated receptor (PPAR)α [15,73]. In this sense, the reduction in serum levels of adiponectin, characteristic in obese individuals, accentuates the imbalance in the balance between the production of pro- and anti-inflammatory adipokines secreted by adipose tissue, minimizing the hepatoprotective and anti-inflammatory effects and representing an independent risk factor for MAFLD and liver dysfunctions, such as NASH. Thus, obesity is closely related to reduced adiponectin levels, as it reduces insulin sensitivity and increases liver fat levels. Furthermore, patients with NASH can show dysregulated postprandial glucose and lipid homeostasis, presenting a greater expression of AdipoR2 in liver tissue [43,61,74,82,141,142].

In summary, adiponectin plays a crucial role in the pathophysiology of insulin resistance. The levels are inversely related to insulin resistance and metabolic syndrome. In this sense, insulin resistance causes abnormalities in lipid storage and lipolysis in insulin-sensitive tissues, which can increase the flow of free fatty acids from adipose tissue to the liver and, consequently, cause steatosis [9,57,143,144,145].

The anti-inflammatory effects of adiponectin are due to the upregulation in stimulating the secretion of anti-inflammatory cytokines, such as IL-10, the inhibition of Nuclear Factor-κβ (NFκβ), and the release of TNF-α and IL-6. It also has antisteatotic action, induced by weight loss, associated with the oxidation of fatty acids in skeletal muscle tissue and the liver, and antifibrotic action, acting to reduce hepatic gluconeogenesis and maintain stellate cells in their quiescent state in the liver. This fact explains its in vitro hypoglycemic effect. Moreover, adiponectin has thermogenic effects and induces fatty oxidation in the liver and skeletal muscle [43,146].

The antisteatotic role occurs through 5-AMP kinase by inhibiting Acetyl-CoA decarboxylase (ACC) and fat synthase. It also reduces the influx of fatty acids into the liver, thus playing an important role in preventing the progression of MAFLD [74,141,147].

Leptin, a product of the ob gene, is a peptide produced by differentiated adipocytes and is related to CNS signaling regarding energy stores in adipose tissue, crossing the blood–brain barrier to bind to its receptor (ObR or LepR), located in the hypothalamus, through two types of neurons in the hypothalamic arcuate nucleus (NHA)—the one that expresses pro-opiomelanocortin (POMC) and the one that expresses cocaine and amphetamine by regulating of the enzyme transcriptase (CART) so that the brain directs adjustments necessary for balance between energy expenditure and consumption, which is linked to the inhibition of food intake. Furthermore, leptin activity in the hypothalamus is positively modulated by insulin, and vice versa, and its circulating levels are directly proportional to adipose tissue reserves. Thus, in obesity, increased adiposity induces hyperleptinemia. This condition can play a role in saturating hormonal receptors and altering blood–brain permeability, failing communication between adipose tissue and the CNS [57,137,147].

Furthermore, hyperleptinemia is related to effects antagonistic to those of adiponectin, stimulating the production of pro-inflammatory mediators such as IL-6 and TNF-α, reducing adiponectin concentration, and decreasing lipolysis in adipose tissue. Thus, leptin acts as a pro-fibrogenic adipokine, as it positively regulates Collagen α1 and increases the differentiation of HSCs, which expresses leptin receptors (LepR), a factor that leads to the establishment of liver fibrosis. Furthermore, it also positively regulates the expression of CD14 in Kupffer cells—endotoxic bacterial lipopolysaccharide (LPS) receptor, which, when positive, causes greater sensitization of cells to harmful stimuli and, consequently, more significant oxidative stress, progression of steatohepatitis and fibrosis [41,56]. In summary, it is possible to point out that the link between leptin and the inflammatory component of MAFLD is summarized in the discovery that leptin increases pro-inflammatory and pro-fibrotic effects [147,148,149,150,151].

Omentin is a hormone associated with increased insulin sensitivity and glucose absorption, with anti-atherosclerotic implications and protection against liver damage. In this sense, its levels are inversely related to obesity, hyperglycemia, inflammation, insulin resistance, and cardiovascular complications. For example, in patients with insulin resistance (diagnosed with DM2), the omentin serum level is reduced, favoring MAFLD/NASH development [143,147,152]. Studies have shown that besides the above actions, omentin can show antioxidation effects and regulate apoptosis. Reduced levels are associated with metabolic syndrome and its comorbidities, such as hypertension, obesity, DM2, and carotid atherosclerosis. A meta-analysis showed that lower omentin levels are related to the occurrence and progression of MAFLD. Clinical studies demonstrated that serum or visceral adipose omentin levels were significantly reduced in subjects with DM2, metabolic syndrome, obesity, and NAFLD. For these reasons, omentin could be used as a non-invasive diagnostic biomarker for MAFLD [133,153,154]. 

Resistin is produced by macrophages and monocytes of adipose tissue and is expressed in abundance by patients with MAFLD, specifically in white adipose tissue. Its secretion is strongly related to insulin resistance linked to obesity, as its levels increase in genetic or diet-induced obesity due to its effects on glucose metabolism, which are antagonistic to those of insulin, increasing hepatic glycogenesis [30]. The expression of resistin mRNA is intense in the liver of patients with MAFLD. This adipokine desensitizes the cells of fat, muscular, and hepatic tissues to insulin, increasing hepatic resistance to insulin, contributing to the endogenous production of glucose, and generating a pro-inflammatory environment. Resistin expression is possibly associated with the gravity of inflammation and hepatic fibrosis. It participates in hepatic fibrinogen through its pro-inflammatory action. CRP is directly connected with the level of circulant resistin, given that it stimulates the expression level of TNF-α, IL6, and IL-12. The circulant levels of TNF-α and IL-6 are higher in patients with MAFLD. In addition, it regulates the expression of TNF-α and IL-1β through the mitogen-activated protein kinase (MEK) and Extracellular-signal-regulated kinase (ERK), inhibiting some microRNAs (miRNAs) [155,156,157].

Vaspin, produced by visceral adipose tissue, due to the inhibitory action of kallikrein 7 protease, responsible for insulin degradation, favors such sensitivity to this hormone, reduces the synthesis of pro-inflammatory cytokines, and protects vascular tissue from possible damage due to fat accumulation [143].

Vaspin is a protein of the serine protease inhibitor family, which is expressed in visceral adipose tissue. In this regard, serum levels of vaspin increase in type 2 diabetes and obesity. It promotes impairment of insulin sensitivity. However, it decreases with diabetes regression, weight loss, and increased insulin sensitivity [158]. Furthermore, insulin-sensitizing adipokine increases metabolic diseases to compensate for insulin resistance and inflammatory complications, contributing to the development of MAFLD [159].

Nicotinamide phosphoribosyltransferase (NAMPT), also called pre-B cell colony enhancement factor (PBEF) or extracellular visfatin, acts mainly as an inducer of the production of pro-inflammatory cytokines and, therefore, has been associated with metabolic and inflammatory disorders [135]. It is produced in adipose tissue by macrophages and adipocytes and, outside of it, by hepatocytes and neutrophils. It helps store triacylglycerols and favors the pro-inflammatory scenario, promoting the dysfunction of pancreatic β cells and the production and release of TNF-α, VCAM-1, and IL-6 by adipose tissue [43,147]. Visfatin expression is regulated by various cytokines such as TNF-α, IL-6, and lipopolysaccharide, which are known to promote insulin resistance [15]. Furthermore, an increase in visfatin levels has been found to be associated with atherosclerotic disease and coronary artery disease. These conditions are among the leading causes of mortality in NAFLD [160]. The insulin-mimetic actions of visfatin are related to the stimulation of glucose uptake by insulin-sensitive cells (adipocytes and myocytes) and the inhibition of glucose release from liver cells [161].

Meteorin-like hormone (Metrnl) is an adipokine mainly synthesized by white adipose tissue that has a crucial role in insulin sensitization, lipid metabolism, energy expenditure, inflammation, and neural development. It can also regulate white adipose browning. The serum levels of Metrnl are reduced in MAFLD patients, suggesting it has a protective role against the development of MAFLD [139].

### 3.2. Myokines

Myokines are produced by skeletal muscle in response to muscle contraction and act in an inhibitory way against the harmful effects of pro-inflammatory adipokines [162,163]. Irisin, active in inducing the differentiation of white adipose tissue (WAT) into brown adipose tissue (BAT), suppresses lipogenesis and cholesterol synthesis, optimizing lipid oxidation and, consequently, lipid homeostasis [69,164]. Myostatin and Mionectin act in the progression of lipid uptake and deposition in the liver, in the propensity for the pro-inflammatory scenario, and in the inhibition of antioxidant compounds [43]. IL-6, in turn, when originating from muscle contraction in a pathway independent of TNF-α, has an anti-inflammatory effect, stimulating the production of acute phase cytokines with an anti-inflammatory profile; thus, when released by the muscle tissue, it promotes lipolysis, glycogen degradation, and the inhibition of TNF-α release [43,138,165]. 

### 3.3. Hepatokines

Finally, when studying hepatokines, it is clear that among their important roles in participating in the pathophysiology of MAFLD, Angiopoietin-Like 4 (ANGPTL 4) stands out, which is released by the liver during physical exercise. Inhibiting pancreatic lipases promotes less fat absorption in obese patients, reducing liver fat concentration. Leukocyte Cell-Derived Chemotaxin 2 (LECT 2), a neutrophil chemotactic organokine directly associated with metabolic stress, compromises insulin signal transduction, especially in myocytes, and increases the production of inflammatory cytokines. In addition, it is known that LECT 2 interacts with several receptor signaling pathways in different diseases, including hepatic, renal, and pulmonary amyloidosis, MAFLD and insulin resistance, liver fibrosis, and cirrhosis [34,166,167].

Furthermore, high levels of Selenoprotein P (SeP) are found in patients with metabolic diseases, positively correlating with insulin resistance. Fetuin-A stimulates the sensitization of TLR-4 in adipocytes and macrophages to promote the pro-inflammatory scenario in conjunction with FFAs, directing IR, the suppression of adiponectin synthesis by adipocytes, and the induction of pancreatic β-cell toxicity; thus, its levels are shown to be elevated in obesity in correlation with MAFLD [151,168].

Sexual Hormone-Binding Globulin (SHBG), secreted by the liver, transports sex steroids to their target tissues. Emerging evidence from epidemiological studies has suggested that SHBG is increasingly recognized as a hepatokine involved in the occurrence and development of MAFLD [169]. Due to this, SHBG is present at reduced rates in individuals with hepatic steatosis. This occurs since lipogenesis, the main pathway that results in the accumulation of ILH (intrahepatic lipids), reduces the expression of hepatocyte nuclear factor 4α—responsible for SHBG transcription—resulting in a reduction in its expression [170,171]. Concerning lipid metabolism, the overexpression of SHBG protects against the development of MAFLD, inhibiting hepatic lipogenesis by controlling the main lipogenic enzymes. In this sense, the genetic tendency to high levels of transcription of the hepatokine SHBG was causally correlated with a lower risk of developing MAFLD [138,172]. Therefore, fat accumulation reduces the production of SHBG, increasing hepatic lipogenesis and exacerbating the development of MAFLD [40,43,173].

The stellakines, substances produced by hepatic stellate cells, also play a pivotal role in liver disease. They include Amyloid Beta Precursor Protein, C-C motif chemokine ligand 2 (CCl2), C-C motif chemokine ligand 11 (CCL11), Colony-stimulating factor 1 (CSF1), Connective tissue growth factor (CTGF), C-X-C motif chemokine ligand 1 (CXCL1), C-X-C motif chemokine ligand (CXCL)10, 14, and 16, Growth arrest-specific gene 6 (GAS6), Netrin 1 (NTN1), Periostin (POSTN), and Wnt Family Member 4 (WNT4). In general, these mediators have actions related to the oxidative stress response of endothelial cells, inflammation, activation of macrophages, monocytes, Kupffer cells, modulation of cytokine and chemokine mediated signaling pathways, adhesion of hepatic stellate cells and endothelial cells, recruitment of fibrocytes, and initiation of chemoattraction of cells [174,175].

Hepatokines are related to the development of MAFLD and NASH through signaling pathways such as adipogenesis, fibrogenesis, pregnane X receptor (PXR)/RXR, farnesoid x receptor (FXR)/retinoic X receptor (RXR), hepatic stellate cell activation, liver X receptor (LXR)/RXR, NF-κB, PPAR, PPARα/retinoic acid receptor alpha, AMP-activated protein kinase (AMPK), and DM2 [174].

Thus, it is concluded that the low chronic inflammatory level in adipose tissue, perpetuated due to the accumulation of fat—lipogenesis—which contributes to the accentuation of obesity, contributes to the establishment of the pathogenesis of metabolic disorders, such as MAFLD [176,177]. In this sense, the inflammatory process in adipose tissue is characterized by the infiltration of macrophages and activation of inflammatory pathways mediated mainly by NF-κB and by adipokines, hepatokines, and myokines by endocrine, autocrine, and paracrine pathways, correlating with high serum levels of pro-inflammatory adipokines, such as leptin and resistin, low concentrations of anti-inflammatory adipokines, such as adiponectin, vaspin, and omentin, and high levels of hepatokines (fetuin-A) and pro-inflammatory myokines (irisin and IL-6), constituting the profile of organokines in the context of obesity [178,179].

### 3.4. Osteokines

Bone Morphogenetic Protein (BMP) stands out among osteokines. It is a member of the TGF-β family responsible for regulating processes such as organogenesis, embryonic development patterns, and the production of white and brown adipocytes. Furthermore, TGF-β signaling is a central pathway for the progression of liver diseases. The BMP-4 subtype presents evidence of a pro-fibrotic role and is secreted by adipocytes, stimulates adipogenesis, and directs pre-adipocytes toward the brown adipocyte phenotype. In obesity, preadipocytes are resistant to this subtype, which may contribute to obesity-related diseases such as MAFLD [168,180,181,182].

Osteocalcin, synthesized and released mainly by osteoblasts and subsequently activated by osteoclasts in bone resorption, stimulates the consumption of glucose and FFA, promoting the expression of fatty acid transporters, stimulating β-oxidation and the translocation of GLUT4 to the plasma membrane. This scenario provides an anti-inflammatory environment, contributing to a reduction in visceral fat. GluOC (uncarboxylated osteocalcin) treatment increases the hepatic insulin signaling pathway, inhibits gluconeogenesis, and promotes glycogen synthesis to improve hyperglycemia. Furthermore, it is related to the stimulation of β-oxidation of fatty acids, which, in turn, prevents hepatic lipid synthesis from improving fatty liver and hypertriglyceridemia, therefore resulting in a de-inflamed environment [22,183,184,185,186,187].

Table 2 shows some characteristics of adipokines, myokines, osteokines, and hepatokines involved in the pathogenesis of MAFLD.

## 4. Advances in Clinical Trials and Therapeutic Options for MAFLD

Currently, non-drug treatment for MAFLD is based on changing eating habits, aiming to reduce visceral lipid accumulation, combined with the initiation of physical activity, responsible for reducing the inflammatory markers characteristic of the pathophysiology of the disease [188,189]. This therapeutic option can be combined with prescription medication to regulate glucose and lipid metabolism, minimizing inflammation and liver damage. According to the American Association for the Study of Liver Diseases (AASLD), the use of GLP-1 receptor agonists, such as Liraglutide, Semaglutide, Dulaglutide, and Exenatide, is recommended for patients with type 2 diabetes mellitus (DM2) and obesity for the treatment of MAFLD to reduce body weight and insulin resistance [50,190,191,192].

Furthermore, sodium–glucose cotransporter-2 (SGLT2) inhibitors, such as Canagliflozin, Dapagliflozin, Empagliflozin, and Ertugliflozin, comprise drugs with a satisfactory therapeutic response in the treatment of MAFLD, as they act to increase urinary glucose excretion, acting as an anti-hyperglycemic drug approved for DM2 and favoring a reduction in the risk of cardiovascular events. Thus, the promising effects of GLP-1 receptor agonists and SGLT2 inhibitors, combined with lifestyle changes, although they offer an optimized approach to reducing NASH, also represent an option for minimizing MAFLD [34,50].

On the other hand, in the case of non-diabetic individuals with MAFLD, the prescription of Pioglitazone, a gamma receptor agonist, showed the regression of hepatic steatosis, lobular inflammation, and hepatocellular ballooning, an improvement in IR and liver enzyme levels, and a reduction in liver damage. However, these changes were followed by the adverse effect of body weight gain [193].

Some studies indicate that the antioxidant effect of vitamin E promotes the minimization of liver decompensation and, therefore, eliminates the need for transplantation in patients with MAFLD [194]. Because of their intervention in the hepatic metabolism of lipids and carbohydrates, in insulin and REDOX signaling, and the regression of the inflammatory process, the administration of Sirtuins (SIRTs) is being evaluated as an additional option in the treatment of MAFLD. SIRTs constitute a family of seven members (SIRT1-7) with distinct cellular locations linked to various cellular processes. SIRT1, in humans with MAFLD, presents reduced regulation associated with increased expression of lipogenic proteins, such as Sterol Regulatory Element Binding Protein 1 (SREBP1), Acetyl-CoA Carboxylase (ACC), and Fas Cell Surface Death Receptor (FAS). Next, the lack of SIRT1 catalytic activity promotes the excretion of free FAs from mesenteric adipose tissue, aggravating MAFLD [164]. Thus, SIRT1 levels are reduced in obese patients and obese patients with severe hepatic steatosis compared with obese patients with mild hepatic steatosis. SIRT3, in turn, improves mitochondrial function, minimizing the negative effects of MAFLD through the regulation of β-oxidation, ketogenesis, mitophagy, and the antioxidant response system [193].

Finally, a surgical option—bariatric—is considered when the patient does not improve with changes in diet and exercise [195]. Figure 4 summarizes the risk factors for MAFLD and the possibilities of treatment.

### Therapeutic Perspectives for MAFLD: The Role of AdipoRon

AdipoRon—a reduced active synthetic molecule—consists of an adiponectin receptor agonist (AdipoR1 and AdipoR2), which is active in enhancing the oxidation of fatty acids and reducing oxidative stress in skeletal striated muscles and in the hepatic store, demonstrating its anti-lipotoxic action and consequent protection against excessive accumulation of fatty acids and subsequent mitochondrial dysfunction in the liver. It is the pioneer with such active oral pharmacodynamics [196,197]. Furthermore, scientific evidence elucidates its anti-inflammatory effect in several models, including acute hepatitis, liver fibrosis, nephropathy, and cardiac inflammation [198,199].

In this sense, Adiponectin Receptor Agonist (ADP355) is a short active synthetic adiponectin-mimetic peptide, altered through the insertion of unnatural amino acids and labeled by the potential to reestablish regular adiponectin activity. Therefore, its action is expected to mitigate harmful plasma changes and the opposite conditions linked to fatty liver. Adiponectin is not directly used because it has a diverse range of protein structures expressed in its chemical structure. This interferes with obtaining reproducible results both in laboratory environments (in vitro) and in living organisms (in vivo), in addition to encouraging the margin of errors and side reactions after its administration [34].

In treatments with AdipoRon, because of its antiproliferative and anticancer properties, not only a reduction in basal mitochondrial respiration and maximum cell respiration but also an attenuation of proton leakage is observed. In this scenario, cancer cells, as a compensatory response to the defective production of mitochondrial ATP, increase anaerobic glycolysis, consume a greater amount of glucose, and increase the synthesis of lactic acid. Consequently, glycolysis inhibitors are potential targets to improve the effectiveness of AdipoRon. Thus, in addition to the aforementioned antitumor properties mediated by AdipoRon in neoplasms, there is a simultaneous inhibition of angiogenesis in specific cells, a factor that causes a shortage of nutrients and a drop in oxygen supply in tumor cells [196].

Furthermore, the activation of AMPK, observed in response to AdipoRon, is correlated with the suppression of insulin resistance and glucose intolerance, highlighting an additional metabolic therapeutic potential for AdipoRon. Furthermore, as an inhibitor of anabolic processes and an activator of catabolic processes, such as lipid breakdown, AdipoRon is capable of suppressing proliferation and, therefore, inducing an anti-cancer response [200].

Regarding the genetic cycle, administering specific dosages of AdipoRon impacts cell growth. In this context, changes in cell cycle speed are detected, as AdipoRon causes a G0/G1 intensification and a depletion in the S phase. On the other hand, no changes in the biodynamics of the G2/M phases appear to occur in the same cells treated with AdipoRon. These findings reinforce AdipoRon as an antiproliferative compound in specific malignant processes and support its effectiveness in discouraging cell cycle progression [201].

## 5. Ferroptosis

Ferroptosis is defined as the uncontrolled accumulation of lipid peroxides—a chain reaction mediated by radicals—which leads to the incorporation of molecular oxygen into lipids. In this sense, such chemical reactions can be spontaneous or catalyzed stereospecific by lipoxygenase (LOX) enzymes. Thus, the binding of the LOX15 complex (ALOX15, 15-LOX) with the phosphatidylethanolamine (PE) binding protein (PEBP1) promotes ferroptosis in some contexts [202,203,204,205].

Therefore, as iron chelators, such as deferoxamine, act to block ferroptosis, this condition is iron-dependent, with excess metal responsible for sensitizing numerous cell types to the process. Therefore, maintaining serum iron homeostasis is essential for protecting cells against ferroptosis [206,207].

Regarding MAFLD, the role of ferroptosis is linked to the high presence of fat in hepatocytes, obesity, and insulin resistance, a typical characteristic in patients with MAFLD that causes an imbalance in iron concentration [77].

When lipid peroxidation occurs, ROS increase, which increases oxidative stress, progressing the development of MAFLD. Among the different aldehydes that can be formed as secondary products during lipid peroxidation, malondialdehyde and 4-hydroxynonenal (4-HNE) stand out, both being associated with different stages of MAFLD [208]. In turn, arachidonic acid and adrenic acid-containing phosphatidylethanolamines (PEs), vulnerable to ROS attack, are the main substrates for lipid peroxidation; such long-chain polyunsaturated fatty acids are preferentially catalyzed to their acyl-CoA esters by long-chain acyl-CoA synthetase family member 4 (ACSL4), re-acylated into lysophospholipids by lysophosphatylcholine acyltransferase 3 (LPCAT3), and subsequently oxidized by lipoxygenase, resulting in membrane rupture and ferroptotic cell death [209].

This scenario causes the death of hepatocytes, reflected by higher levels of AST and ALT; thus, hyperoxidized peroxiredoxin 3 (PRDX3), a ferroptotic marker, is observed in livers exposed to these conditions, confirming that ferroptosis contributes to liver injury in MAFLD. Additionally, in MAFLD, hepatic levels of hyperoxidized PRDX3—a marker of ferroptosis—are positively correlated with serum levels of AST and ALT, which are markers of liver damage. Therefore, inhibition of ferroptosis is essential to prevent the development and progression of the disease [210].

### 5.1. Ferroptosis and the Intestinal Microbiota

Recently, an association between the gut microbiota and cellular ferroptosis was discovered. Dietary Fe^3+^ is absorbed by duodenal intestinal epithelial cells and reduced to Fe^2+^ by the divalent metal ion transporter-1 (DMT1). In turn, Fe^2+^ absorbed into the blood is oxidized to Fe^3+^ by ceruloplasmin, bound by transferrin, and transported to tissues. However, due to the first-pass effect of hepatic portal circulation, iron exposure in the liver is much greater than in other tissues, resulting in liver damage and various complications. Thus, changes in iron metabolism, resulting in fluctuations in its serum level, can potentially influence the occurrence of ferroptosis [77,211,212].

In this sense, the microbiota of the human gastrointestinal tract—formed by commensal bacteria—establishes a competitive relationship with the host for the use of iron for its physiological activity through the expression of the protein FeoB (Ferrous iron transport protein B). The ideal condition is that the host and commensal microorganisms acquire sufficient iron for their life processes without suffering induced death due to excess iron. Therefore, if there is a deficiency in the gastrointestinal tract, ferroptosis may be triggered, worsening the MAFLD condition [213].

### 5.2. Ferroptosis and Organokines

Recent studies highlight that lipid hydroperoxide dependent on glutathione peroxidase 4 (GPX4) has the potential to use glutathione (GSH) as a cofactor for the conversion of lipid peroxides in non-toxic lipid alcohols, thus protecting cells from lipid peroxidation. Therefore, inhibition of GPX4 is capable of inducing the development of ferroptosis [214,215,216].

In contrast, inhibition of GPX4 in some cell lines may not trigger ferroptosis. In these cases, the coenzyme Q oxidoreductase FSP1 (ferroptosis suppressor protein 1)—independent of GSH—acts in parallel with GPX4, representing another primary regulator of ferroptosis. FSP1, located in the plasma membrane, consumes NADH/NADPH as an oxidoreductase, called Ubiquinone, CoQ10 dependent on NADH/NADPH, reducing it to CoQ10H2—Ubiquinol. Thus, CoQ10 is a lipophilic molecule, present mainly in the inner membrane of mitochondria, while CoQ10H2 acts as a lipophilic antioxidant to capture free radicals, preventing peroxidation [217].

Likewise, the mitochondrial protein Dihydroorolated dehydrogenase acts to inhibit ferroptosis through the reduction of ubiquinone to ubiquinol within the inner mitochondrial membrane. Another recently studied mechanism, independent of GPX4, demonstrates that cells can use nitric oxide (NO) for self-defense against ferroptosis. It is known that M1 macrophages express the iNOS member of the NOS family (nitric oxide synthetases). Due to their high concentrations of NO, combined with stimulation with lipopolysaccharide and IFN-γ, M1 macrophage acts in the self-suppression of ferroptosis [218].

## 6. MAFLD and the Development of Hepatocellular Carcinoma

Liver cancer ranks third among deaths from neoplasms in the world, preceded by lung and colorectal neoplasms. Among liver neoplasms, hepatocellular carcinoma (HCC) stands out, which constitutes 75% of liver tumors [219].

Concerning patients with MAFLD, several factors influence the development of HCC, such as liver cirrhosis, DM, advanced age, the male sex, alcohol consumption, and smoking, that is, the progression of the condition. The emergence of neoplasia is driven by metabolic imbalance and lipotoxicity resulting from lipid overload of hepatocytes and oxidative stress, while genetic markers, intestinal dysbiosis, and alcohol and tobacco abuse can interact as risk modifiers for such cellular alteration. Genetically, the variant in the PNPLA3 gene is the most favorable in linking fatty liver to HCC, compromising the adequate processing of fat by liver cells, in addition to the TM6SF2 gene polymorphism also being associated with increased liver fat content in MAFLD, fibrosis, advanced liver disease, and cirrhosis [220].

Depletion of p53—a tumor suppressor protein—favors the regression of the differentiation of mature hepatocytes into progenitor cells. The alteration in the formation of HCC, followed by a genetic mutation in the Wnt and Notch signaling pathways, acts mainly in the development of embryonic cells, coordinating cell differentiation, cell proliferation, and apoptosis [221].

Likewise, metabolic imbalance, represented by IR, is the primary pathogenic event associated with the development of hepatic steatosis and HCC. Thus, this condition leads to the synthesis and activity of insulin-like growth factor-1, inhibiting cell proliferation and apoptosis, which increases the risk of hepatocellular carcinogenesis. Likewise, lipotoxicity is the deregulation of intracellular lipid components, promoting their accumulation in cells, subsequent damage, and possible cell death. Furthermore, increased intestinal permeability, excessive growth of intestinal bacteria, and serum endotoxin are observed in MAFLD and HCC. In parallel, the induction of TLR-2 is caused by endotoxemia, which promotes the production of prostaglandin E through the activation of the cyclooxygenase-2 pathway, suppressing antitumor immunity, inhibiting the production of antitumor cytokines from cells liver immune systems, and, consequently, progressing HCC [222].

Finally, oxidative stress, resulting from the imbalance between the excessive formation of ROS and/or reactive nitrogen species and a reduction in antioxidant defenses, predisposes cell death and possible tissue damage since ROS contain unpaired electrons, decisive in the abnormal growth and development of cancer cells [223].

## 7. Conclusions

MAFLD represents a multifaceted condition intricately woven into the fabric of metabolic dysfunctions. This comprehensive review offered insights into the intricate relationship between MAFLD and a spectrum of metabolic disorders. The comorbidities such as T2DM, MetS, and IR are particularly noteworthy, which compound the clinical complexity and exacerbate liver lipogenesis imbalance and systemic inflammation. Moreover, our examination underscored the pivotal role of the brain–gut–liver axis in orchestrating the pathophysiology of MAFLD. We illuminate the profound impact of inflammatory processes, cellular alterations within hepatocytes and stellate cells, and lifestyle factors like hypercaloric diets and sedentarism on disease prognosis.

Furthermore, our exploration delved into the crucial significance of organokines—adipokines, myokines, osteokines, and hepatokines—in modulating the physiological responses underlying MAFLD. Adipokines emerge as critical regulators of insulin sensitivity and inflammation, while myokines contribute indispensably to adipose tissue homeostasis, both playing pivotal roles in mitigating the progression of MAFLD. Additionally, hepatokines emerge as critical determinants, influencing the disease trajectory by regulating pro-inflammatory and anti-inflammatory responses.

Deepening our comprehension of the intricate molecular mechanisms underpinning MAFLD and its associated comorbidities is imperative. Future research endeavors should prioritize elucidating the clinical implications of organokine modulation and exploring novel therapeutic avenues to target these pathways. By harnessing the potential of organokines, we can chart innovative approaches to manage MAFLD effectively and enhance patient outcomes, thereby advancing the landscape of liver disease management.

## 8. Future Perspectives

Given the substantial impact of organokines on the progression of MAFLD, there is a pressing need for future investigations to focus on pioneering novel therapeutic approaches that target these organokines’ signaling pathways directly. This scenario could involve exploring pharmacological interventions or lifestyle modifications to modulate the secretion or activity of adipokines, myokines, osteokines, and hepatokines, thereby offering potential avenues for managing MAFLD. Furthermore, there is a compelling rationale for further exploration of the complex interplay between the brain–gut–liver axis and organokine signaling in the pathogenesis of MAFLD. Research efforts to elucidate how central and peripheral signals integrate to regulate the secretion and action of organokines could provide a more comprehensive understanding of the underlying disease mechanisms. To comprehensively understand MAFLD pathophysiology and progression, future studies must also prioritize the identification of novel biomarkers associated with organokine dysregulation in MAFLD. These biomarkers could serve as invaluable diagnostic tools for early detection, prognostic indicators for disease progression, and targets for monitoring therapeutic efficacy. 

Investigating organokine replacement therapies beyond adiponectin with AdipORon holds promise for MAFLD prevention and treatment. While adiponectin has been extensively studied for its beneficial effects on metabolic health, other organokines, such as irisin [224] and hepatocyte growth factor (HGF) [225], have also shown potential in preclinical and clinical studies. Irisin, a myokine released during exercise, has been implicated in improving glucose homeostasis [226] and lipid metabolism [227], suggesting its potential as a therapeutic target for MAFLD. Additionally, HGF, primarily produced by stromal cells [228], exhibits potent anti-inflammatory and regenerative properties in the liver [229]. Utilizing organokine-replacement therapies to restore the physiological levels of these molecules could mitigate metabolic dysfunction, attenuate liver inflammation, and promote hepatic regeneration in MAFLD patients. Further research is warranted to elucidate the safety, efficacy, and optimal dosing regimens of these organokines in clinical settings, with the ultimate goal of developing novel therapeutic approaches for MAFLD management. Similarly, myostatin inhibition has been shown to enhance muscle mass and insulin sensitivity while reducing hepatic lipid accumulation in animal models of obesity and diabetes [230].

Scientists and laboratory companies must collaborate to effectively mitigate the costs of measuring organokines in blood samples. By pooling their expertise and resources, scientists can contribute valuable insights into developing innovative measurement techniques and assays for organokines. Simultaneously, laboratory companies can leverage their technological capabilities and infrastructure to streamline the organokine analysis process, thus optimizing efficiency and reducing costs. Through this collaborative effort, both parties can work synergistically to advance the field of organokine research while making these crucial measurements more accessible and cost-effective, ultimately benefiting scientific progress and healthcare outcomes.

Investigating the genetic underpinnings of MAFLD and its interaction with organokine signaling also presents a promising avenue for future research. Genetic studies have begun to uncover essential susceptibility genes and polymorphisms associated with MAFLD development and progression. By integrating genetic data with our understanding of organokine biology, researchers can elucidate how genetic variants influence the secretion, activity, and downstream effects of adipokines, myokines, osteokines, and hepatokines in MAFLD pathogenesis. Furthermore, exploring gene–environment interactions may provide insights into how genetic predispositions interact with lifestyle factors to modulate organokine expression and contribute to disease risk. Leveraging advances in genomic technologies, such as genome-wide association studies and transcriptomic profiling, holds the potential to identify novel genetic markers and pathways implicated in MAFLD pathophysiology. Ultimately, integrating genetic research with the study of organokine signaling pathways may offer new perspectives on individualized risk stratification, precision medicine approaches, and targeted therapeutic interventions for MAFLD.

Finally, the potential synergistic effects of combining medication with organokine replacement therapy represent a promising avenue for the treatment of MAFLD. Drugs targeting specific metabolic pathways, such as insulin sensitizers or lipid-lowering agents, may complement the actions of organokines in addressing critical aspects of MAFLD pathophysiology. For example, combining organokine replacement therapy with insulin sensitizers like metformin or thiazolidinediones could amplify the beneficial effects on glucose metabolism and hepatic insulin sensitivity. Similarly, the co-administration of lipid-lowering agents such as statins or fibrates may enhance the reduction in hepatic lipid accumulation when combined with organokines known to modulate lipid metabolism. Furthermore, medications targeting inflammation or fibrosis, often observed in advanced stages of MAFLD, could synergize with organokines possessing anti-inflammatory or regenerative properties. By harnessing the complementary mechanisms of action of medications and organokines, combination therapies have the potential to provide more comprehensive and efficacious treatment strategies for MAFLD, offering new hope for patients burdened by this complex metabolic disorder. However, rigorous preclinical and clinical studies are warranted to elucidate such combination approaches’ safety, efficacy, and optimal dosing regimens in managing MAFLD.

In conclusion, it is imperative to bridge the gap between basic research on organokine signaling and its clinical application in managing MAFLD. Translational research endeavors should be geared toward developing evidence-based interventions that leverage our evolving understanding of organokine biology to enhance patient outcomes and mitigate the burden of MAFLD-related complications. These forward-looking research perspectives aim to build upon the foundation laid by current studies and address critical gaps in our understanding of MAFLD pathophysiology and treatment. By centering on organokines and their intricate involvement in the brain–gut–liver axis, researchers can propel our knowledge forward and pave the way for more targeted and productive therapeutic strategies for MAFLD.

## Figures and Tables

**Figure 1 ijms-25-03694-f001:**
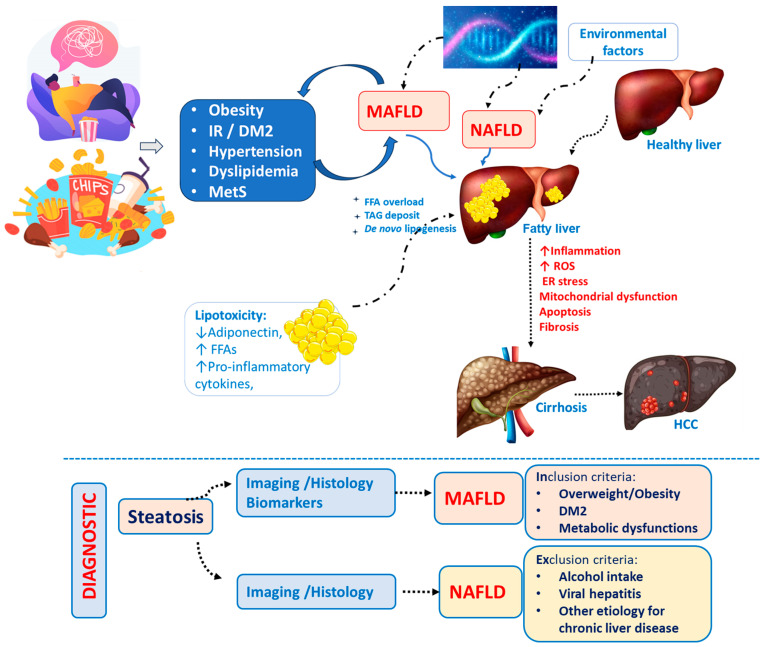
Differences between the diagnosis and definition of nonalcoholic fatty liver disease (NAFLD) and metabolic-associated fatty liver disease (MAFLD) stem from environmental factors and genetics, which contribute to increased lipid deposition in liver tissue, leading to inflammation and oxidative stress. In NAFLD, the presence of steatosis is primarily related to genetics but not to other causes beyond insulin resistance. DM2: type 2 diabetes mellitus; ER: endoplasmic reticulum; FFAs: free fatty acids; HCC: hepatocellular carcinoma; MetS: metabolic syndrome; ROS: reactive oxygen species; TAG: triglycerides.

**Figure 2 ijms-25-03694-f002:**
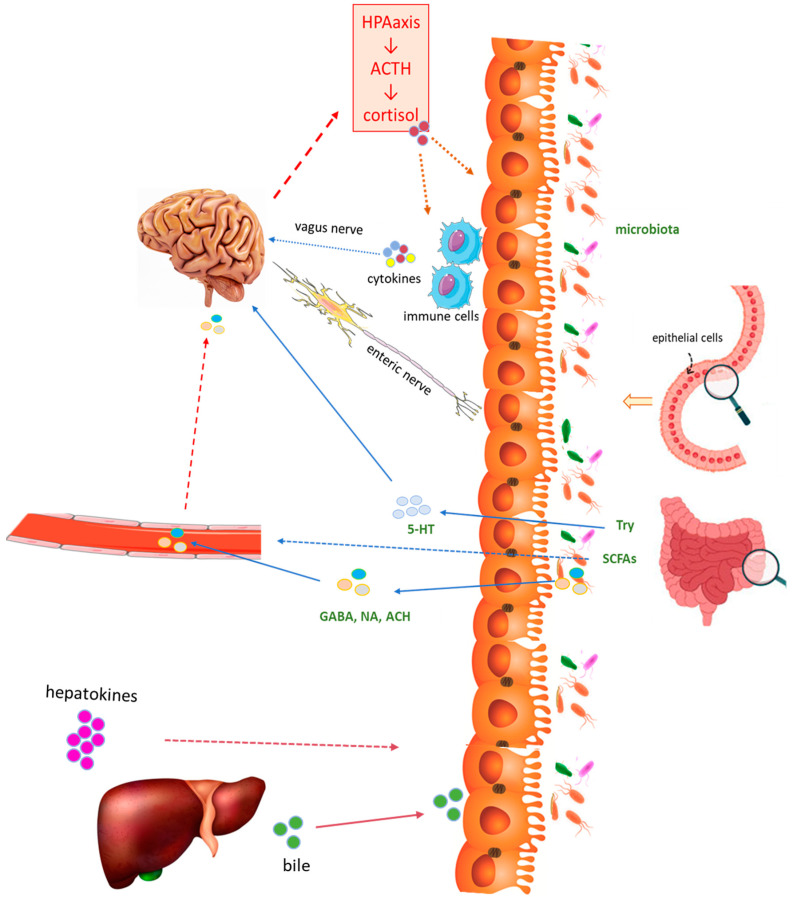
The complex interaction of the brain–gut–liver axis. In the figure, bidirectional communication pathways occur among the brain, gut, and liver. These pathways include endocrine, immune, and neurological connections. Microbiota can release tryptophan (Trp)/5-hydroxytryptamine, short-chain fatty acids (SCFAs), Noradrenalin (NA), and Gamma-aminobutyric acid (GABA), which have systemic actions. The hypothalamic–pituitary–adrenal (HPA) axis in the endocrine pathways can respond to the environment or endogenous stress, producing corticotropin-releasing hormone (CRH) and resulting in the release of cortisol. This stressor substance interferes with microbiome, neuronal, and physiological processes. In the immune pathways, the gut can interact with the brain through signaling biomarkers such as pro-cytokines, influencing neuroimmune responses. Microbiota products can modulate immune cells and regulate inflammation both locally and systemically. The liver is connected to the enteric system thanks to the portal vein and can liberate hepatokines, which also have systemic actions. The colored arrows only indicate the interrelationships between one organ/system and another.

**Figure 3 ijms-25-03694-f003:**
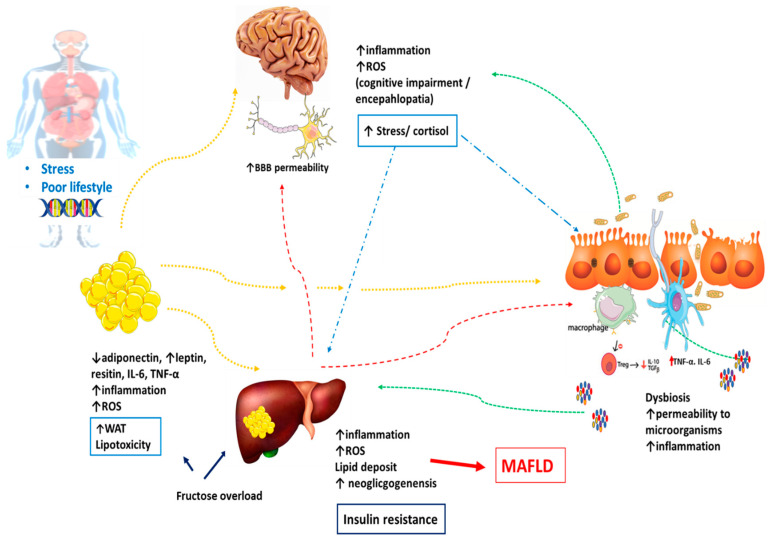
The relationships of metabolic-associated fatty liver disease (MAFLD). Increased gut permeability triggers immune system activation and the release of pro-inflammatory cytokines that contribute to brain inflammation and the production of stressors, which is related to increased inflammation and oxidative stress in the liver. The products of the low-grade inflammation of adipose tissue aggravate this pro-inflammatory scenario, which contributes to systemic inflammation, leading to cognitive impairment. IL-6: Interleukin-6; ROS: reactive oxygen species; TNF-α: Tumor Necrosis Factor-α. The colored arrows only indicate the interrelationships between one organ/system and another.

**Figure 4 ijms-25-03694-f004:**
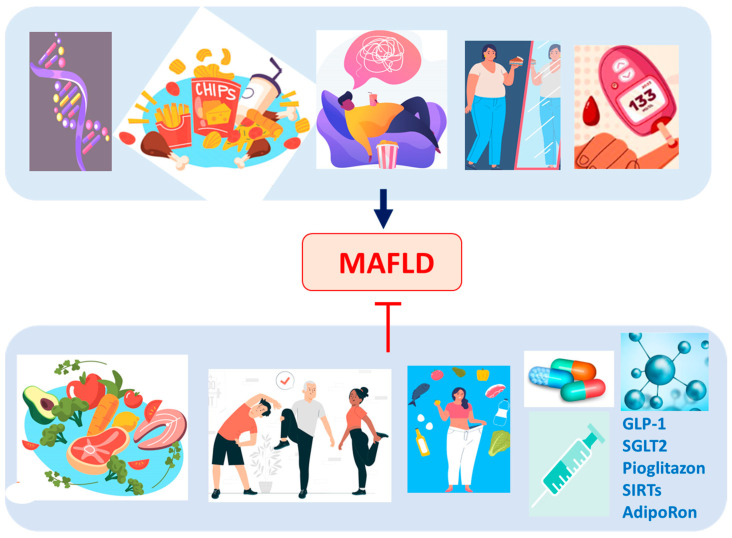
Risk factors and the possibilities for treatment. Lifestyle modification (a change from a Western diet to healthier foods, the practice of physical exercise, and a reduction in body weight and waist circumference) associated with medicines such as Glucagon-like peptide-1 receptor agonists (GLP-1), sodium–glucose cotransporter-2 (SGLT2), pioglitazone, Sirtuins (SIRTs), and antioxidants such as vitamin E can reducerisk factors and can reduce inflammation, oxidative stress, lipid accumulation, steatosis, and cirrhosis.

**Table 1 ijms-25-03694-t001:** Mechanisms of oxidative stress and their relationship with MAFLD.

Free Radical	Source	Relationship with MAFLD	References
ROS	Mitochondria and metabolism	Stimulating conformational changes in the mitochondrial structure; Aggravation of lipid accumulation in hepatocytes; facilitation of lipid droplet remodeling; Induction of inflammatory cytokines, which contribute to liver inflammation and the development of carcinomas; Compromising the integrity of mitochondria-associated membranes (MAMs).	[55,60,77,80]
RNS	Metabolic processes	Compromising the integrity of mitochondria-associated membranes (MAMs); Oxidation of fatty acids.	[22,80,81]

**Table 2 ijms-25-03694-t002:** Main characteristics of some adipokines, myokines, osteokines, and hepatokines involved in the pathogenesis of MAFLD.

Classification	Organokine	Production	Action	Role in MAFLD	Ref.
	Adiponectin	Adipose tissue	Anti-inflammatory: Secretion of cytokines, such as IL-10; blocking NF-κβ activation; inhibition of the release of TNF-α and IL-6 Antisteatostatic Antifibrotic weight loss: Inhibition of hepatic glycogenesis	Protective adiponectin, its levels are inversely proportional to insulin resistance	[69,137,147]
	Leptin	Adipose tissue	Pro-inflammatory: Secretion of IL-6 and TNF-α MAFLD progression Reduces adiponectin concentration Reduces adipose tissue lipolysis	Related to insulin resistance or failure of the antisteatotic effect	[69,137,147,168]
ADIPOKINE	Omentin		Increases peripheral sensitivity to insulin and glucose absorption Protection against atherosclerosis and liver implications	Inversely related to obesity, hyperglycemia, inflammation, and insulin resistance	[145,147]
	Resistin	Adipose tissue macrophages and monocytes	Reduces the number of mitochondria Elevates lipid accumulation	Expressed abundantly by patients with MAFLD	[147,168]
	Vaspin	Visceral dipole tissue	Increased insulin sensitivity: Inhibitory action of kallikrein 7 protease, responsible for insulin degradation Anti-inflammatory: Reduced production of pro-inflammatory cytokines	Protects vascular tissue from possible damage due to fat accumulation	[147]
	Visfatin	Macrophages and adipocytes of adipose tissue and hepatocytes and neutrophils	Helps in the storage of triacylglycerols: Dysfunction of pancreatic beta cells Production and release of TNF-α, VCAM-1, and IL-6	Related to a pro-inflammatory scenario	[145,147,171]
MIOKINE	Irisin	Adipose tissue	Differentiation of white adipose tissue (WAT) into brown adipose tissue (BAT) Suppresses lipogenesis and cholesterol synthesis: Optimization of lipid oxidation	Improves lipid homeostasis	[163,168]
	Myostatin e Mionectin		Pro-inflammatory Inhibits antioxidant compounds	Progression of lipid uptake and deposition in the liver	[163]
OSTEOKINE	BMP-4	Adipocytes	Regulation of metabolic processes: Adipogenesis, targeting preadipocytes toward the brown phenotype	In obesity, pre-adipocytes are resistant to this subtype of BMP, which may contribute to diseases related to this condition	[69,180]
	Osteocalcin	Osteoblasts	Stimulation of glucose and fatty acid consumption: Expression of fatty acid transporters	Anti-inflammatory scenario that contributes to a reduction in visceral fat	[168]
HEPATOKINE	ANGPTL-4	Liver	Inhibition of pancreatic lipases	Related to less fat absorption, and in obese patients, there is a decrease in their liver levels	[168,178]
	LECT-2		Neutrophil chemotaxis Weight gain Detects the development of hepatic steatosis	Associated with metabolic stress; it compromises insulin signal transduction in addition to increasing the secretion of inflammatory cytokines	[166,178]
	SHBG	Liver	Transport of sex steroids Overexpression protects against the development of MAFLD: Inhibition of hepatic lipogenesis by control of lipogenic enzymes	Decreased rate in patients with hepatic steatosis. Fat accumulation reduces SHBG production by increasing hepatic lipogenesis and exacerbating the development of MAFLD	[169,171,178]

ANGPTL-4: Angiopoietin-Like 4; BMP: Bone Morphogenetic Protein; IL: Interleukin; MAFLD: metabolic-associated fatty liver disease; LECT 2: Leukocyte Cell-Derived Chemotaxin; NF-κβ: Nuclear Factor κβ; SHBG: Sexual Hormone-Binding Globulin; TNF-α: Tumor Necrosis Factor-α; VCAM: Vascular cell adhesion protein 1.

## Data Availability

Not applicable.

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
