# Peer review of "Underlying Mechanisms behind the Brain–Gut–Liver Axis and Metabolic-Associated Fatty Liver Disease (MAFLD): An Update"

_ijms, 2024, doi:10.3390/ijms25073694_

Round 1
Reviewer 1 Report
Comments and Suggestions for Authors
The review entitled "Underlying Mechanisms behind Brain-Gut-Liver Axis and Metabolic-Associated Fatty Liver Disease (MAFLD): An Update" is very well conceived, updated data regarding the pathogenesis of MAFLD and the Brain-Gut-Liver Axis, as well and organokines. Also, the Figures are very clear and illustrative.
Minor revision:
In the paragraph: 2.4. Oxidative Stress and MAFLD - to add a table that will contain all mechanisms of oxidative/nitrosative stress, i.e. sources of ROS/RNS in MAFLD with appropriate references.
Author Response
The review entitled "Underlying Mechanisms behind Brain-Gut-Liver Axis and Metabolic-Associated Fatty Liver Disease (MAFLD): An Update" is very well conceived, updated data regarding the pathogenesis of MAFLD and the Brain-Gut-Liver Axis, as well and organokines. Also, the Figures are very clear and illustrative.
Response: Dear doctor, thank you very much for your time reviewing this manuscript. Thank you for your kind comments.
Minor revision:
In the paragraph: 2.4. Oxidative Stress and MAFLD - to add a table that will contain all mechanisms of oxidative/nitrosative stress, i.e. sources of ROS/RNS in MAFLD with appropriate references.
Response: Dear doctor, we appreciated your suggestion and included the table in section 2.4, page 7, lines 318-319.
Dear doctor, thank you again for reviewing and improving the quality of this manuscript. With best regards.
Reviewer 2 Report
Comments and Suggestions for Authors
MAFLD is of growing interest to clinicians and researchers due to its important contribution to the pattern of liver disease. In this regard, the presented review is qualitative and informative.
Comments:
1. The figures need to be improved. For example, colored arrows are drawn. It is not clear what the colors of the arrows mean and why some arrows are solid and some are dotted. Does it make any difference?
2. The article has two sections 4: Advances in Clinical Trials and Therapeutic Options for MAFLD and Ferroptosis. The article needs to be better structured so that there is a more consistent description.
3. The article uses the terms intestinal microbiota and gut microbiotaa, as well as brain-gut-liver axis and cerebral-hepatic-intestinal axis. Is there an implied difference between the terms?
4. Check the language in which lines 574-579 are written
5. Since Brain-Gut-Liver Axis is the main axis in the manuscript, it is recommended to describe it in more detail. It is also recommended to describe the role of SCFA in more detail.
Comments on the Quality of English LanguageCheck the language in which lines 574-579 are written
Author Response
MAFLD is of growing interest to clinicians and researchers due to its important contribution to the pattern of liver disease. In this regard, the presented review is qualitative and informative.
Response: Dear doctor, thank you very much for your time correcting this manuscript. Thank you for your comments and suggestions.
Comments:
- The figures need to be improved. For example, colored arrows are drawn. It is not clear what the colors of the arrows mean and why some arrows are solid and some are dotted. Does it make any difference?
Response: Dear doctor. You are correct! We improved the legend of the figures.
- The article has two sections 4: Advances in Clinical Trials and Therapeutic Options for MAFLD and Ferroptosis. The article needs to be better structured so that there is a more consistent description.
Response:
- The article uses the terms intestinal microbiota and gut microbiota, as well as brain-gut-liver axis and cerebral-hepatic-intestinal axis. Is there an implied difference between the terms?
Response: You are right. We corrected these problems along with the text. We also changed cerebral-hepatic-intestinal axis for brain-gut-liver axis.
- Check the language in which lines 574-579 are written
Response: Thank you for your correction. We modified the sentence.
- Since Brain-Gut-Liver Axis is the main axis in the manuscript, it is recommended to describe it in more detail. It is also recommended to describe the role of SCFA in more detail.
Response: Dear reviewer, these important suggestions were included. Please see page 2, lines 49-60 and 64-69.
Dear doctor, thank you again for reviewing and improving the quality of this manuscript.
Reviewer 3 Report
Comments and Suggestions for Authors
Authors provide the review on “Underlying Mechanisms behind Brain-Gut-Liver Axis and Metabolic-Associated Fatty Liver Disease (MAFLD): An Update”.
This review covers the areas associated with MAFLD and brain-gut-liver axis.
• I would like to suggest adding MAPkinase for the completeness in the mitochondrial ROS section. Activation of mitogen-activated kinases (MAPK) cascade plays important role in the mitochondrial ROS production. The nutritional and metabolic stress causes sustained c-Jun N-terminal kinase (JNK) activation in the liver. (Ref.#) The sustained activation of JNK via the self-amplified feed-forward activation loop is found to play a key role in several disease models such as acute liver injury, ischemia-reperfusion injury in heart, cardiotoxicity, neurotoxicity, and fatty liver diseases.
(Ref.# Hepatology. 2021 PMID: 34331779; PMCID: PMC8639630)
• In the brain-gut-liver axis section, I suggest including Aβ metabolism in the liver-brain axis. Aβ metabolism takes place in the liver and dis-regulation of Aβ metabolism effects the liver and the brain.
• And also, additional including the bile acid role. The high fat and cholesterol diet (Western diet) changes the bile acid profile in the liver, and that associates with neurological diseases, Alzheimer's and Parkinson's Diseases.
• Missing Choline and choline-related metabolites, trimethylamine (TMA), TMA is a product of choline metabolism by the intestinal microbiota. Choline is required for the very low-density lipoprotein (VLDL) formation in the liver. In case of choline deficiency VLDL accumulates in the liver and causes increased lipid peroxidation and reactive oxidative stress formation in the liver.
Author Response
This review covers the areas associated with MAFLD and brain-gut-liver axis.
Response: Dear doctor, thank you very much for your time correcting this manuscript. Thank you for your comments and suggestions.
- I would like to suggest adding MAPkinase for the completeness in the mitochondrial ROS section. Activation of mitogen-activated kinases (MAPK) cascade plays important role in the mitochondrial ROS production. The nutritional and metabolic stress causes sustained c-Jun N-terminal kinase (JNK) activation in the liver. (Ref.#) The sustained activation of JNK via the self-amplified feed-forward activation loop is found to play a key role in several disease models such as acute liver injury, ischemia-reperfusion injury in heart, cardiotoxicity, neurotoxicity, and fatty liver diseases.
(Ref.# Hepatology. 2021 PMID: 34331779; PMCID: PMC8639630)
Response: Thank you for this crucial suggestion. We included MAPK information and include the suggested references. Please see page 5, lines 212-233.
- In the brain-gut-liver axis section, I suggest including Aβ metabolism in the liver-brain axis. Aβ metabolism takes place in the liver and dis-regulation of Aβ metabolism effects the liver and the brain.
- And also, additional including the bile acid role. The high fat and cholesterol diet (Western diet) changes the bile acid profile in the liver, and that associates with neurological diseases, Alzheimer's and Parkinson's Diseases.
Response: Dear Doctor, we included these important comments. Please see page 10, lines 372-449.
- Missing Choline and choline-related metabolites, trimethylamine (TMA), TMA is a product of choline metabolism by the intestinal microbiota. Choline is required for the very low-density lipoprotein (VLDL) formation in the liver. In case of choline deficiency VLDL accumulates in the liver and causes increased lipid peroxidation and reactive oxidative stress formation in the liver.
Response: We also included these important comments. Please see page 11, lines 449-454.
Dear doctor, thank you again for reviewing and improving the quality of this manuscript.
With our best regards.
Round 2
Reviewer 2 Report
Comments and Suggestions for Authors
The authors made corrections to the text, which improved the quality of the article. However, two sections 4 were retained in the manuscript. It is necessary to correct the numbering of the sections.